# Knowledge about Neonatal Screening among Postpartum Women and Complexity Level of Birthing Facilities

**DOI:** 10.3390/ijns5010008

**Published:** 2019-01-22

**Authors:** Mariana F. Guimarães, Fernanda A. Rabelo, Israel Figueiredo

**Affiliations:** Departamento Materno-Infantil, Faculdade de Medicina, Hospital Universitário Antonio Pedro, Universidade Federal Fluminense, Niterói 24033-900, Brasil

**Keywords:** confirmatory factor analysis, neonatal screening, postpartum period, knowledge

## Abstract

Background: To ascertain the degree of knowledge of postpartum women about important aspects related to the neonatal screening process and whether differences of opinion exist between those who deliver in low-complexity versus high-complexity health facilities (low-risk versus high-risk pregnancies, respectively). Methods: This was a prospective, cross-sectional, questionnaire-based study. The sample consisted of postpartum women recruited from 2013 to 2015 at public maternity hospitals in the city of Niterói, Brazil. Participants were divided into two groups and completed a questionnaire consisting of Likert-scored items. Continuous variables were analyzed with the Mann-Whitney test, and categorical variables, with Fisher’s test. A confirmatory factor analysis of participants’ answers was performed. Results: Of 188 women enrolled, 54 (28.7%) had incomplete elementary education; 119 (62.2%) had attended more than six antenatal care visits. The mean age was 25.57 years. Nearly all women (*n* = 179, 95.2%) were roomed-in with their infants. Knowledge of neonatal screening was very similar in the high-complexity and low-complexity groups. Divergences were limited to items regarding the risks of neonatal screening. Conclusions: The degree of knowledge among postpartum women was similar among high- and low-complexity facilities. Those who attended high-complexity facilities had longer hospital stays and greater adherence to ethical issues regarding neonatal screening.

## 1. Introduction

Early detection of the diseases covered by neonatal screening (NS) is a longstanding and extremely important practice. Since Garrod’s first description of alkaptonuria in 1902 and the introduction of post-natal screening for phenylketonuria in the 1960s, preventive, predictive, personalized, and participatory (so-called “P4”) medicine has been a reality for physicians who manage patients with hereditary metabolic diseases. [1]

In addition to the traditional tests performed after birth, which can detect conditions difficult to identify by physical examination alone, several improved or novel prenatal and perinatal screening procedures have been developed, and major advances in neonatal diagnosis and prognosis have been achieved [2,3,4,5,6].

However, this air of modernity stands in stark contrast to the harsh reality of perinatal and neonatal care in many settings. Mandatory basic screening for metabolic diseases has not been implemented universally; it is lacking in countries as large as China [7] and Iran [8]. In others, implementation of these basic screening tests has taken many years to reach a reasonable level of population coverage due to a series of factors—including failure of many facilities to perform screening tests, inadequate referral of newborns to facilities not equipped to collect samples and perform tests, and poorly trained professionals, as seen most recently in Brazil, where coverage stands at approximately 70.8% (95%CI 69.0–72.7) [9]. Conversely, as of 2011, a comprehensive neonatal screening panel covering at least 26 disorders was offered in all 50 U.S. states [10].

This lack of full population coverage in many countries, the low priority given to dissemination of information about screening, and the limited investment therein create systemic ignorance on the topic of NS. Despite these gaps in implementation of even the most basic NS procedures, as technology advances, new tests [1] are being imposed, with their attendant costs [10]; perversely, while investments are made in the development and implementation of these new tests, large portions of the world’s neonatal population remain uncovered by basic screening panels and their parents uninformed about the benefits of NS.

Several issues involved in metabolic screening are widely known and have been the subject of research, such as detection of benign conditions and complex results without a clear diagnosis. However, one issue that has received little attention is that this development of a wide-ranging armamentarium of NS procedures has taken place with no regard for the opinion of women during the prenatal or puerperal period. Few studies [8] have set out to understand the awareness and knowledge of mothers and perinatal care providers in relation to neonatal screening tests, especially regarding the ethics of the classic heel-stick test.

In addition to these aspects, there is no clear picture of the relationship between the complexity level of the birth facility and the qualitative and quantitative information received by the parents before, during and after delivery. It is assumed that mothers who go through a longer hospital stay have a greater chance of passively receiving additional information about NS.

Within the present context, this study aimed to qualitatively define the degree of knowledge among postpartum women about important aspects related to the neonatal screening process and to ascertain whether differences exist in the opinions of women who delivered in low- versus high-complexity facilities, with an emphasis on the ethical aspects of screening.

## 2. Materials and Methods

### 2.1. Study Design

This was a prospective, cross-sectional, mixed-methods (qualitative-quantitative), questionnaire-based study. The sample consisted of postpartum women recruited from 2013 to 2015. Participation was voluntary, and all provided written informed consent. The present study was approved (5 January 2014) by the Fluminense Federal University Research Ethics Committee (opinion no. 23136714.7.0000.5243).

### 2.2. Setting and Participants

This study was carried out in three public maternity hospitals in the city of Niterói, state of Rio de Janeiro, Brazil, two of which are classified as high-complexity facilities and the third as a low-complexity facility. These were the most important maternity units in the city during the study period, accounting for the majority of hospital births in the region. All three units were able to provide a complete range of obstetric care, including cesarean sections and other invasive obstetric procedures, and staffed by obstetricians, anesthesiologists, and neonatologists to provide care to mother and child at all stages of the birth process. The low-complexity facility has an open-door, walk-in policy, while access to the high-complexity units was restricted to patients with known high-risk pregnancies or obstetric complications referred by primary- or secondary-care units of the public health network. It should be noted that the low-complexity unit was endowed with advanced life support resources (for both mother and newborn) and trained staff, which allowed provision of adequate support until transfer to a tertiary (high-complexity) unit in case of complications.

Postpartum women who were interested in the subject were included in the study regardless of whether they had received adequate antenatal care or specific information about NS and the diseases it detects. To maximize questionnaire coverage, the data collection sites and dates were staggered. Participants were enrolled by random selection of chart numbers, using a computer-generated random numbers list. Upon completion of the questionnaire, each participant received an informative brochure on the topic. Three women who refused to complete the questionnaire were excluded from analysis.

Written informed consent was obtained from all participants, both in the pilot study (*n* = 25, see below) and in the final sample (*n* = 227), not only for collection of data from medical records but also to authorize the administration of specific items in the study questionnaire. Consent for screening itself was not obtained in the maternity ward, but rather at a designated specimen collection point set up at another municipal health facility to which mother-child dyads were referred.

### 2.3. Sample Size and Data Source

A pilot test was carried out in a sample of 25 postpartum women to evaluate the data collection instrument. This pilot stage lasted 6 months, and led to modifications of the content of the questionnaire. The final version of the instrument was administered over a period of approximately one and a half years. The final database comprised 227 respondents. It should be noted that, although formal technical guidance from the Brazilian Ministry of Health is available [11], the prenatal information system is not consistent across the country, since families can choose to follow up at public or private facilities in the most diverse levels of complexity of the health system. The questionnaire was administered by two of the authors, who had been previously trained and always worked in pairs to minimize intra and inter-observer bias. For data analysis, 39 questionnaires were excluded because respondents claimed to have no knowledge of the subject. Thus, the final sample comprised 188 women who responded to the statements presented by the investigators.

### 2.4. Questionnaire

The first part of the questionnaire collected general data to establish a profile of participants, such as the hospital in which data were collected, the participant’s municipality of residence, age, and educational attainment, and variables related to pregnancy, delivery, and neonatal history.

The second part consisted of 36 closed-ended (multiple-choice) statements, each with five possible answers. These statements sought to elicit information [12] on Concepts of neonatal screening: (1) Knowledge about prevention; (2) Benefits of testing; (3) Treatment to prevent health problems; (4) Presence of clinical signs during the first weeks of life; (5) Whether the test should be performed at the maternity hospital; (6) Where parents should go to have the test performed if it was not performed at the place of delivery). Collection of blood samples: (7) The heel-stick test; (8) The exact timing of collection; (9) Problems associated with early collection; (10) Collection before 48 h of life; (11) Collection after the 7th day of life; (12) Optimal timing of collection between the 3rd and 7th days of life). Diseases covered by neonatal screening: (13) The four diseases initially covered by the test; (14) Phenylketonuria; (15) Hypothyroidism; (16) Sickle-cell anemia; (17) Cystic fibrosis; (18) Treatment of these diseases and possibility of cure; (19) Whether other screening tests exist; (20) Whether diagnostic tests should be performed even when there is no specific therapy). The local neonatal screening system: (21) Tests to prevent future complications; (22) What happens in the event of a positive test and specialized follow-up; (23) Active case-finding system in the event of positive results; (24) Detection of other carriers in the family; (25) That a single sample could elucidate more than 40 diseases; (26) That each state can have its own screening protocol. Risks of neonatal screening: (27) Need for confirmation; (28) Risk of failing to identify some false negatives; (29) Odds of false-positive and family stress; (30) Possibility of detecting false paternity) and questions regarding the Ethics of neonatal screening: (31) Informed consent requirement; (32) Knowledge of the destination of collected blood; (33) The nature of the results; (34) The need for a second blood draw; (35) Right of refusal; (36) Universal nature of the tests.

Each statement was followed by a five-item Likert-type scale, which offered the following possible positive or negative response options: (1) Completely disagree; (2) Partially disagree; (3) Neutral; (4) Partially agree; (5) Completely agree. These data were used as the basic substrate for confirmatory modeling through principal components analysis.

### 2.5. Database

There were no missing data. Thus, no methods such as expectation/maximization, imputation, or others had to be used to fill in missing values.

### 2.6. Variables and Groups

Overall, 47 variables were selected and analyzed. The nine categorical variables were: place of birth (dichotomous: low-complexity = 0, high-complexity = 1), educational attainment (six categories: illiterate = 1; incomplete primary education = 2; completed primary education = 3; incomplete secondary education = 4; completed secondary education = 5; any higher education = 6), antenatal care (three categories: none = 1; up to 5 visits = 2; > 6 visits = 3), mode of delivery (vaginal = 0, cesarean = 1), need for neonatal resuscitation (dichotomous: no = 0, yes = 1) and hospitalization setting (rooming-in = 1; high-dependency unit = 2; intensive care unit = 3). Variables concerning provision of information during antenatal care (yes = 0, no = 1) and at the maternity ward (yes = 0, no = 1) were also assessed.

The two continuous variables were respondent age (numerical, years), and length of hospital stay (numerical, days). In addition to the aforementioned variables, the 36 Likert-scored, qualitative, and ordinal questionnaire items were included in the study.

### 2.7. Statistical Methods

Data were processed in IBM^®^ SPSS^®^ Statistics for Windows, Version 24.0 (IBM Corp, Armonk, NY, USA). Statistical significance was accepted at *p* < 0.05 for all tests.

Categorical variables were reported as absolute and relative frequencies. The complexity of the birthing facility (low = 0; high = 1) was cross-tabulated with the other categorical variables, and tests for association were performed using the Pearson chi-square method, taking into account Fisher’s exact test.

Continuous variables were expressed as means, medians, confidence intervals, and standard deviations. Variables were tested for normality of distribution to decide whether parametric (ANOVA) or nonparametric (Mann–Whitney and Kruskal–Wallis tests) methods should be used. Age and length of hospital stay were also used as dependent variables, while information on complexity level of birthing facility, educational attainment, antenatal care, mode of delivery, need for neonatal resuscitation, and setting of hospitalization were used as predictors to construct a Generalized Linear Model (GzLM). A log-γ distribution was chosen for both the age variable and length of hospital stay, because it provided the best fit, as demonstrated by the lowest Akaike information criterion (AIC) values. Bonferroni’s post-hoc test was used to adjust for multiple comparisons.

Any divergences in response to the 36 questions on neonatal screening were recorded. The high-complexity and low-complexity groups were analyzed separately to test for possible relationships, percent agreement and disagreement, median, and mode in the sample overall and in each group. A confirmatory factor analysis (CFA) of the data collected with the study questionnaire was then performed. According to the hypothesis raised, the observed values (P31 to P36) would form a cluster to explain whether the factor Knowledge of Ethics (latent variable) would be related to the birthing facility (high- vs. low-complexity). A diagram was plotted to evaluate this theory. The comparative fit index (CFI), with a cutoff value of ≥0.95, was used as a measure of adequacy. To ascertain whether the data fit the model, a Root Mean Square Error of Approximation (RMSEA) indicator was applied, expecting values in the range <0.06–0.08. The software settings were also adjusted to improve the reliability of the model by limiting the variance of the latent variable to 1 (mean 0, variance limited to 1–Z-score). The quality of the model and its limitations were discussed. All modeling was carried out in IBM^®^ SPSS^®^ Amos™ 24.0.0 software.

## 3. Results

Of the 188 postpartum women enrolled in the three birthing facilities, 116 (61%) delivered at a high-complexity facility. Most had completed secondary education (*n* = 63, 33.7%), followed by incomplete elementary education (*n* = 54, 28.7%). Seven (3.7%) had a higher education and one (0.5%) was illiterate. Overall, 119 women had attended more than six antenatal care visits (62.2%), while seven had not attended any (3.7%). The mean age was 25.57 years (95%CI 23.67–26.47; SD 6.265). In the high-complexity group, most deliveries were by cesarean section (*n* = 69, 59.5%), while in the low-complexity group, vaginal birth predominated (*n* = 38, 52.8%). The overall rate of cesarean delivery was 54.8% (*n* = 103). Most newborns did not require resuscitation in the delivery room (*n* = 162, 86.2%). The vast majority of mothers roomed-in with their infants (*n* = 179; 95.2%), for an average of 3.57 days (95% CI 2.14–4.00, SD 3.017). There were no significant differences (*p* > 0.05, chi-square) in any of these categorical variables between the low-complexity and high-complexity birthing facility groups.

However, women admitted to high-complexity facilities were older on average (mean = 26.15, 95% CI 24.94–27.35, SD 6.560) than those admitted to low-complexity units (mean = 24.64, 95% CI 23.30–25.97, SD 5.680), although the difference was not significant (MW-*p* = 0.149). As expected, there was a significant difference (MW-*p* < 0.001) between length of stay at high-complexity facilities (mean = 4.16 days, 95% CI 3.50–4.82, SD 3.585) compared to low-complexity units (mean = 2.61, 95% CI 2.31–2.91, SD 1.284). The average length of stay was also longer among those who reported having received information at the maternity ward (mean = 4.33, 95% CI 3.12–5.54, SD 2.869) than among those who denied having received any guidance on neonatal screening (mean = 3.46, 95% CI 2.99–3.92, SD 3.030), although the difference was not significant (MW-*p* = 0.082; GzLM, γ distribution *p* = 0.094).

Analysis of responses to the questionnaire showed what the 118 women thought about the topic of interest. Concerning statements related to the *concept* of neonatal screening (Q1 to Q6), there was clear agreement on the importance of prevention (82.5%), the benefits of testing (80.9%), that proper treatment can prevent sequelae or death (48.4%), and whether the child should be taken to another facility for screening (71.3%), while disagreeing that one should wait for symptoms to appear before testing (91.0%); most were neutral as to whether screening should be done at the birthing facility (39.4%).

Regarding *collection of blood samples* (Q7–Q12), there was clear agreement on use of the heel-stick technique (85.7%), that it would be wrong to perform screening as soon as possible after birth (80.9%) and after the first 48 h (64.4%), as well as neutral as to the optimal timing of sample collection, whether on the 7th day (41.0%) or between the 3rd and 7th days (80.9%).

Regarding the *diseases covered* (Q13–Q20), the respondents demonstrated indecision, neutrality, and lack of knowledge about the official neonatal screening program which detects four diseases (98.8%), phenylketonuria (100%), congenital hypothyroidism (98.9%), sickle-cell anemia (97.9%), cystic fibrosis (100%), and whether treatments and cures were available for all of these diseases (37.2%). The respondents agreed that the program should cover the greatest possible number of diseases, including extension of the program to cover other illnesses (95.3%) and/or even for diseases in which the results would be of little or no benefit to those affected (96.3%).

Regarding aspects of the *official screening system* (Q21–Q26), the respondents mostly disagreed that screening alone would be enough to prevent complications in the newborn (69.7%), that the government was right to offer only four tests rather than the 40 diseases covered by neonatal screening in private laboratories (61.2%), and regarding the differences in the number of diseases covered by the screening program in each state (49.5%). They were neutral and displayed a lack of knowledge about the active part of the screening system (active case-finding, treatment, and follow-up of positive cases) (72.9%), and agreed with the need for pediatric follow-up during treatment (94.2%) and for testing other family members for any diseases detected (92.5%).

When questioned about the *risks of screening* (Q27–Q30), they agreed that the test constitutes population-based screening, that positive results should be confirmed (46.8%), that some neonates may not be diagnosed (false negatives) (43.1%), and that a false-positive result may lead to anxiety and worry among family members (43.1%), while they were neutral about the possibility of detecting false paternity during screening (75%).

Finally, when asked about *ethical* issues (Q31–Q36), the vast majority agreed that written informed consent should be required before screening (94.1%), that mothers had to know about the destination and disposition of samples collected for screening (87.3%), that all screening results should be confidential (75.0%), that information should be provided on which tests are offered in case of any modifications to the official program (98.4%), and that screening should be universal, using the same disease panels across all regions of the country (91.0%). They disagreed only regarding the right of parents to refuse screening, in whole or in part (86.2%).

Both groups showed very similar responses to the questionnaire (Table 1). Divergences were limited to statements regarding the risks of neonatal screening (Q27 = need for confirmation of results, Q28 = odds of not identifying some cases, Q29 = odds of false-positive results and family stress); women who delivered at high-complexity facilities were neutral in relation to these risks of neonatal screening (Table 1).

The six statements (Q31-Q36) were pooled (*y* = independent variables), with their Likert responses, to explain the Ethics factor (*x* = latent variable; dependent variable) through CFA of both groups (high-complexity and low-complexity). As the value of the variance of the latent variable was limited to 1 (mean = 0, variance limited to 1–Z-score), the correlation coefficients between the errors were very small, and the coefficients of factor loads (correlation between the variable and the factor) were high (Table 2).

The CFI, was 1.00 and 0.954, while model fit, evaluated by RMSEA, was 0.000 and 0.058 in the low-complexity and high-complexity groups, respectively. The factor load (β value), with its respective *p*-value, showed better convergence in the group of women who had delivered at high-complexity facilities (Figure 1).

## 4. Discussion

The results of this study demonstrate the difficulty of understanding the topic of interest, due to its intrinsic complexity. The respondents’ answers evinced a mix of ignorance and uncertainty. From the outset, there was clear agreement on the importance of neonatal screening and the need for continuity and expansion of the current screening system, but, at the same time, the respondents were somewhat neutral to where screening should be performed and were unaware of the active case-finding protocol in the event of positive results.

Although the majority of patients reported that they had not received information on neonatal screening in the antenatal or nursery settings, their answers suggested that they had some familiarity with the exact timing of the heel-stick test, thus ruling out the possibility of very early or late sample collection. Perhaps an important addition to the NS system would be the figure of an advocate assigned to the mother who could participate in antenatal consultations and provide guidance on screening tests and their results in the post-immediate delivery period, reinforcing the information already offered during antenatal care.

Although they agreed on the importance of testing other family members and providing pediatric follow-up in the event of positive screening, the respondents were unaware of the active case-finding component of the system and were displeased upon finding out that their children could be receiving screening for more than 40 diseases currently collected to screen for only four. The newborn screening system is generally interpreted as being the entirety of actions which result in good outcomes for screened babies, i.e., antenatal and maternity care, birthing facilities, laboratory facilities, pediatricians, and the family. One important component of this system is the panel of screening tests itself. However, new tests are being added to screening panels without the funding or infrastructure needed to provide adequate care, follow-up, and clinical services to newborns and their families in the event of a positive result [13]; only one component of the system is being strengthened, with little attention to other equally or even more important ones. It is noteworthy that many health professionals still also have little or marginal understanding of neonatal screening.

The respondents agreed, although not very emphatically, that repeated testing should be performed in case of suspicion of a false-positive or false-negative result. They were neutral to the possibility of a false-positive result causing anxiety and concern to the parents. Despite the combination of neutral position and ignorance seen in the respondents’ answers to the questionnaire, studies such as this one, which contribute to the knowledge base on the psychosocial impact of medical surveillance triggered by diagnostic uncertainty, should be encouraged, and further assessment and support for these families must be ensured [14]. The present findings also highlight the need to improve the flow of information and the mechanisms whereby parents are notified of test results and supported. Recommendations should be made to address some of the challenges mentioned by participants in order to further mitigate the impact of positive results on families [15].

Regarding the *ethical issues* raised herein, the majority of respondents agreed with the need to obtain consent before the test; most wanted to know what would be done with the blood samples collected; most expected that results be kept confidential; and most expected that, should any changes be made to screening panels in future, they be informed of such changes before any new tests are carried out. They also agreed to new blood collection to confirm suspicious or inconclusive results, although this practice is not generally considered an ethical issue. It was gratifying and reassuring to find that nearly all respondents believed parents should not have the right to refuse newborn screening. One of the major ethical problems of neonatal screening is that parents may receive several types of results that lead to prognostic uncertainty and/or an unexpected probability or diagnosis that may or may not be related to the reason for which the tests were originally performed. This is why excess testing during screening is ethically problematic: the information these tests provide is often neither useful nor relevant to decision making [16]. In the event of such a situation, an emotionally favorable approach should be sought [17]. The main suggestion would be that expanding the provision of information could work not only to alert parents but also to help them, based on individual and collective philosophical definitions, and allow expansion of women’s and parents’ participation in public policy planning to improve newborn screening. This broad discussion is necessary and urgent, since screening based on molecular biology tests (rather than conventional metabolic assays) will soon engender great debate in the ethical field as it becomes more widespread.

Limitations of this study include the relatively small sample size, which prevented us from obtaining results that could generate a systemic impact on the world population, despite the inherent difficulty and complexity of the topic of interest. The relevance of the qualitative findings related to ethics and the satisfactory convergence of statistical data allow us to detect a quantitative trend toward a greater understanding of ethics among those who delivered at high-complexity facilities, highlighting the possibility that this study alone was insufficient to detect significant differences in this dimension. Another limitation of the study was the inability to verify the impact of the screening system on the provision of knowledge to the family. As data collection with the questionnaire happened before the screening process itself was started with sample collection, it was impracticable to quantify the informative influence of the system on the family. Our analysis was restricted to inquiring whether mothers had received information during the prenatal and postpartum periods. It should also be clear that application of the questionnaire did not delay referral to screening, and that ascertaining whether it might increase acceptance and accomplishment of screening was not within the scope of our proposal. We understand that the topic of maternal/family knowledge regarding neonatal screening (particularly via the heel-stick test) and how it impacts acceptance of screening and follow-up/diagnostic testing is critical to developing and refining effective neonatal screening systems.

## 5. Conclusions

In conclusion, this study showed just how difficult the topic of newborn screening is for laypersons; that the degree of knowledge about important aspects related to the newborn screening process is not consistent among postpartum women; and that the opinions of women who delivered in low- versus high-complexity birthing facilities diverged only regarding the risks of screening. Regarding the ethical aspect of screening, the responses of women who delivered in high-complexity birthing facilities corresponded better to the data reported in the existing literature.

## Figures and Tables

**Figure 1 IJNS-05-00008-f001:**
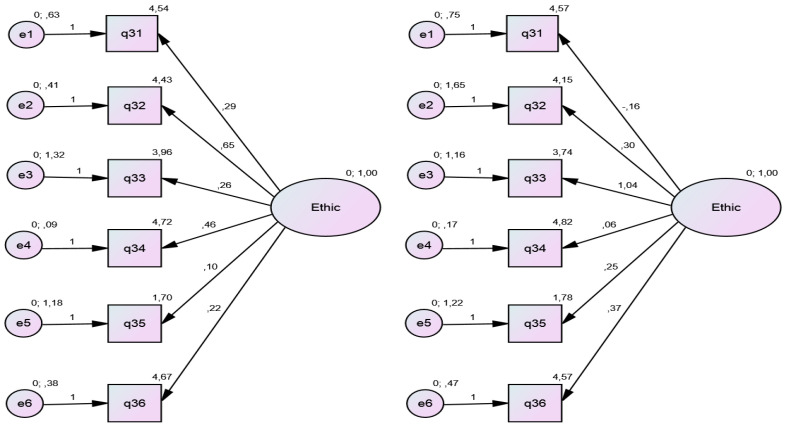
Reflective model—confirmatory factor analysis (CFA) of the *ethics* construct, based on six statements from the knowledge of neonatal screening questionnaire. A. 116 postpartum women who delivered at high-complexity facilities; B. 72 postpartum women who delivered at low-complexity facilities. Ethic—latent variable; Q31–Q36, observed data (six Likert-scored items on ethics of neonatal screening); e, error.

**Table 1 IJNS-05-00008-t001:** Likert rank-order scale of ordinal qualitative measures.

Item	CD		PD		*N*		PA		CA		Median		Mode	
Cplx	High	Low	High	Low	High	Low	High	Low	High	Low	High	Low	High	Low
1	0	1	1	1	18	12	81	43	16	15	4.00	4.00	4	4
2	0	0	0	0	19	17	80	38	17	17	4.00	4.00	4	4
3	0	2	1	7	56	31	53	23	6	9	4.00	3.00	3	3
4	95	51	12	13	1	0	4	3	4	5	1.00	1.00	1	1
5	6	17	23	20	50	24	33	6	4	5	3.00	2.00	3	3
6	2	3	2	2	32	13	57	20	23	34	4.00	4.00	4	5
7	0	0	0	0	15	12	59	37	42	23	4.00	4.00	4	4
8	13	9	66	25	20	14	14	22	3	2	2.00	3.00	2	2
9	1	0	5	3	19	8	75	31	16	30	4.00	4.00	4	4
10	12	28	52	29	49	15	3	0	0	0	2.00	2.00	2	2
11	7	7	30	19	52	25	26	19	1	2	3.00	3.00	3	3
12	0	6	3	2	97	55	16	9	0	0	3.00	3.00	3	3
13	0	0	0	0	116	70	0	2	0	0	3.00	3.00	3	3
14	0	0	0	0	116	72	0	0	0	0	3.00	3.00	3	3
15	0	0	0	0	116	70	0	2	0	0	3.00	3.00	3	3
16	0	0	0	0	114	70	2	2	0	0	3.00	3.00	3	3
17	0	0	0	0	116	72	0	0	0	0	3.00	3.00	3	3
18	4	3	28	18	42	28	35	16	7	7	3.00	3.00	3	3
19	0	0	2	1	2	4	51	23	61	44	5.00	5.00	5	5
20	0	1	1	1	0	4	49	19	66	47	5.00	5.00	5	5
21	17	15	70	29	8	10	21	16	0	2	2.00	2.00	2	2
22	0	0	0	0	6	5	77	42	33	25	4.00	4.00	4	4
23	0	2	3	1	83	48	18	7	12	14	3.00	3.00	3	3
24	1	4	2	1	2	4	51	37	60	26	5.00	4.00	5	4
25	17	21	51	26	15	11	27	13	6	1	2.00	2.00	2	2
26	12	16	43	22	33	16	24	17	4	1	3.00	2.00	2	2
27	3	4	42	20	23	8	40	27	8	13	3.00	4.00	2	4
28	4	4	45	18	22	14	43	31	2	5	3.00	3.50	2	4
29	4	4	44	19	22	14	43	31	2	5	3.00	3.50	2	4
30	3	4	7	16	101	40	5	12	0	0	3.00	3.00	3	3
31	2	3	5	0	0	1	30	17	79	51	5.00	5.00	5	5
32	1	7	9	5	1	1	33	16	72	43	5.00	5.00	5	5
33	4	9	19	13	2	0	44	16	47	34	4.00	4.00	5	5
34	2	0	27	0	0	1	0	11	87	60	5.00	5.00	5	5
35	68	40	33	21	3	1	6	7	6	3	1.00	1.00	1	1
36	0	0	2	2	6	7	20	11	88	52	5.00	5.00	5	5

Cplx, complexity level of birthing facility; *N* = 188; *n*: High-complexity = 116; *n*: low-complexity = 72; CD, completely disagree; PD, partially disagree; N, undecided or neutral; CA, completely agree; PA, partially agree.

**Table 2 IJNS-05-00008-t002:** Confirmatory factor analysis at low-complexity (low) and high-complexity (high) birthing facilities.

			Estimate		S.E.		C.R.		*p*	
			Low	High	Low	High	Low	High	Low	High
q31	<---	Ethics	−0.162	0.294	0.139	0.087	−1.171	3.381	0.242	***
q32	<---	Ethics	0.303	0.646	0.210	0.099	1.444	6.538	0.149	***
q33	<---	Ethics	1.045	0.261	0.372	0.123	2.811	2.130	0.005	0.033
q34	<---	Ethics	0.060	0.464	0.066	0.063	0.909	7.397	0.364	***
q35	<---	Ethics	0.246	0.098	0.180	0.115	1.371	0.855	0.170	0.392
q36	<---	Ethics	0.365	0.216	0.149	0.067	2.445	3.202	0.014	0.001

Estimate, factor load; S.E., standard error; C.R., regression weight estimate; *p* = significance (*** *p* < 0.001).

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
