# Peer review of "Knowledge about Neonatal Screening among Postpartum Women and Complexity Level of Birthing Facilities"

_2409-515X, 2019, doi:10.3390/ijns5010008_

Round 1
Reviewer 1 Report
This is an important study for all programmes where the community is requested to participate and give 'informed' consent, including screening preconception and postpartum. Are they given adequate information, what is adequate and are they really aware of the consequences of testing. In neonatal screening programmes how informed is the consent? and how detailed need the information be? These are open questions and this investigation is important. I felt that some improvement in the presentation is necessary:
The abstract has some unclear points, also unclear in the text. Example is the description of low and high complexity facilities - what is the difference, are patients selected or do they choose according to proximity or according to complications? One can guess the answers but it does make a difference whether for example the mothers in high complexity units are their because of more complications etc Please make it more clear to readers who do not know the Brazilian Health facilities. Also it is stated that 28.7% had completed primary education, but in the text most had completed secondary education, which seems to say that most have a good level of education and only a minority are poorly educated,
In the introduction lines 47-48 it is stated that it has taken many years to 'reach a level of population uptake' but no statement of what it was sin the past so that the level of 70.8% was reached. The reference given [9] does not from the title really refer to neonatal screening, even though this might be my mistake since I could not access the text.
In methodology I wonder whether the 39 ladies who had no knowledge should be excluded from the analysis. Did they sign 'informed consent' before a sample was taken from their baby?
the questionnaire was carefully prepared but if it is a tool that may be copied by other centres (this does not seem to be the intention) it may need to be simplified.
Line 166: 'one' was illiterate
In the discussion/conclusion it may be useful to point out that many health professionals also have little or marginal understanding of neonatal screening (not NS - LINE 50)
The women who delivered in high-complexity facilities should be more clearly described
References - is the format used (first author and then et al - acceptable to the journal?
Author Response
Point 1: “This is an important study for all programmes where the community is requested to participate and give 'informed' consent, including screening preconception and postpartum. Are they given adequate information, what is adequate and are they really aware of the consequences of testing. In neonatal screening programmes how informed is the consent? and how detailed need the information be? These are open questions and this investigation is important. I felt that some improvement in the presentation is necessary.”
Response 1: We agree with the reviewer’s point about these open questions. It is clear that there is no standardization of the necessary items of information that should be passed on to parents during antenatal care, childbirth itself, and the postpartum period.
In an “open-door” socialized medicine system such as the Brazilian Unified Health System, this issue of quantitative/qualitative information has not been emphasized, at least partly because of the lack of uniformity between the public and private care systems and between low- and high-complexity facilities in many regions of the country. The question of consent both during preconception and postpartum (How are parents informed? How detailed is the information provided? Are they informed of potential complications of the test? The possibility of new diagnostic tests?) has been little debated in our public health system. This work sought to obtain such information in the public health facilities of a municipality with a high Human Development Index and found that, despite the credibility of the system in conducting screening and conducting active case-finding, mothers still had many unanswered questions.
Point 2:“The abstract has some unclear points, also unclear in the text. Example is the description of low and high complexity facilities - what is the difference, are patients selected or do they choose according to proximity or according to complications? One can guess the answers but it does make a difference whether for example the mothers in high complexity units are their because of more complications etc Please make it more clear to readers who do not know the Brazilian Health facilities. Also it is stated that 28.7% had completed primary education, but in the text most had completed secondary education, which seems to say that most have a good level of education and only a minority are poorly educated,”
Response 2: The abstract has been modified to clarify these points, but not much, so as not to increase the number of words past the journal limit. A detailed description of the low- and high-complexity facilities and the differences between them is now provided in the Methods section.
P 1 Abstract l 13-14 and 19-20
Point 3:“In the introduction lines 47-48 it is stated that it has taken many years to 'reach a level of population uptake' but no statement of what it was sin the past so that the level of 70.8% was reached. The reference given [9] does not from the title really refer to neonatal screening, even though this might be my mistake since I could not access the text.”
Response 3: The possible causes of this slow rise in coverage rates have been described elsewhere. Reference 9 (http://dx.doi.org/10.1590/1806-93042016000200005) confirms percentage data reported in the text.
P 2 Introduction l 42-47
Point 4:“In methodology I wonder whether the 39 ladies who had no knowledge should be excluded from the analysis. Did they sign 'informed consent' before a sample was taken from their baby?”
Response 4: All 252 women (pilot study and final sample) signed the consent form prior to collection of data from their respective medical records (first part of the survey) and prior to administration of the questionnaire.
All birth facilities in our municipality send mother-child dyads to a designated sample collection point at another unit in the public health system for blood sampling and basic screening. Thus, consent for screening itself was not a part of this study; our informed consent applied solely to medical records data and the questionnaire.
These details have been added to the Methods section.
P 3 Methods l 110-114
Point 5: “the questionnaire was carefully prepared but if it is a tool that may be copied by other centres (this does not seem to be the intention) it may need to be simplified.”
Response 5: The content of the questionnaire was based on a recent Brazilian review article about neonatal screening (doi:10.2223/JPED.1790), which is cited in the Methods section. At the time of writing, there was no intention of wider application of the instrument.
P 3 Methods l 120
Point 6:“Line 166: 'one' was illiterate”
Response 6: The frequency has been added in parentheses to maintain standardization.
P 5 Results l 195
Point 7:“In the discussion/conclusion it may be useful to point out that many health professionals also have little or marginal understanding of neonatal screening (not NS - LINE 50)”
Response 7: We thank the Reviewer for this suggestion, and have made the necessary modification.
P 9 Discussion l 306-308
Point 8: ‘The women who delivered in high-complexity facilities should be more clearly described”
Response 8: Access to the high-complexity facilities was restricted to patients with known high-risk pregnancies or obstetric complications referred by primary- or secondary-care units of the public health network. A detailed description has been added to the Methods section.
P 2 Methods l 82-90
Point 9“References - is the format used (first author and then et al - acceptable to the journal?”
Response 9: The reference format was double-checked and is according to journal guidelines (EndNote - MDPI.ens).
P 11 and 12 References
Reviewer 2 Report
This is an important topic with the potential to make an important contribution to screening systems worldwide however the impact is significantly reduced by the addition of variables other than women's knowledge and the impact of this on the acceptance of screening and the design of screening systems.
The study focussed on metabolic screening by metabolite analysis of dried blood spot samples. There are sufficient problematic issues in this area that the introduction would be stronger if the focus was on these (eg detection of benign conditions, complex results without a clear diagnosis) rather than on prenatal testing, genetic testing and imaging (l 38-42). There are certainly wide inequities in screening worldwide but the introduction does not make it clear how this relates to maternal knowledge (l 44-47). Or how costs relate (ref 10) to family knowledge or understanding.
The introduction does not give any rationale for the hypothesis that knowledge might be different in high complexity facilities rather than low complexity. There is a high proportion of C-sections in both types of facility - is this usual or would more straightforward deliveries be homebirths? ie how does the population of women studied reflect women giving birth in the region?
#2.3 there is high proportion of women claiming no knowledge of the topic. It would be helpful to explain how women get knowledge about screening as this is may be limited which would in turn limit the understanding of the participants. It would be useful also to suggest in the discussion how this might be improved (of course will be particular to the local healthcare system).
l 166 number illiterate is missing.
1 199-200 please clarify difference between diseases and most diagnoses.
l 206 and elsewhere. The usual meaning of indifferent is lack of interest or caring about a topic (although it does have secondary meanings around not having strong feelings), however neutral might be a better choice (meaning neither agreeing or disagreeing) as it does not have the implication of not caring.
Discussion l 259-270 it is unclear how the difficulties of deciding what disorders are on screening panels and international and national inequity of screening relates to maternal knowledge in this community.
l275-294 it is unclear how new genetic technologies and inadequacy of pain management impact women's knowledge of screening. The potential for screening by genomic methodologies is certainly important as are the limitations of this technology but the relevance to the study is not explained.
l299 the newborn screening system is generally interpreted as being the entirety of actions which result in good outcomes for screened babies ie maternity care, birthing facilities, laboratory, paediatricians, family, rather than a panel of screened disorders.
1305 and earlier - it would be interesting for the readers to know how non-paternity is discovered during the screening process (presumably only when family studies are done after a diagnosis is made) and whether separate informed consent is required at this stage (as it would be in many other places) ie not an important part of the initial stages of screening and diagnosis.
l309-312 is the most important part of the discussion and would be good to see positive suggestions.
l317 recollection to confirm screening results is not generally considered an ethical issue any more than thyroid function tests and imaging ie part of a screening protocol.
l325-327 would be good to see this part clarified.
An important aspect of this type of study is the impact on screening uptake of family knowledge. It would be relevant to know when the questionnaire was administered with respect to offering the test and whether the questionnaire affected uptake and how maternal knowledge (or lack of for the 39 excluded) affected screening acceptance for their babies.
In summary the topic of maternal/family knowledge of bloodspot metabolic screening and how it impacts acceptance of both screening and followup/ diagnostic testing is critical in developing and refining effective newborn screening systems and if this write up was confined to issues around current newborn bloodspot testing and included suggestions for improvement and relevance to screening acceptance it would make a strong contribution to the field.
Author Response
Thanks for the comments and the English language and style have been revised, with the necessary modifications. Also Introduction and conclusions were reinstated.
Point 1:“This is an important topic with the potential to make an important contribution to screening systems worldwide however the impact is significantly reduced by the addition of variables other than women's knowledge and the impact of this on the acceptance of screening and the design of screening systems.”
Response 1: We agree with the reviewer that it would be interesting to focus on the core of the research. The text has been revised to focus on the knowledge and impacts of this knowledge on the acceptance of screening and its design.
Point 2:“The study focussed on metabolic screening by metabolite analysis of dried blood spot samples. There are sufficient problematic issues in this area that the introduction would be stronger if the focus was on these (eg detection of benign conditions, complex results without a clear diagnosis) rather than on prenatal testing, genetic testing and imaging (l 38-42).”
Response 2: We have added information on these known issues of neonatal screening to the introduction, as well as an additional sentence on the novelty of the issues addressed by our study (maternal knowledge and opinions, complexity of birth facility).
P 2 Introduction l 56-58
Point 3:“There are certainly wide inequities in screening worldwide but the introduction does not make it clear how this relates to maternal knowledge (l 44-47).Or how costs relate (ref 10) to family knowledge or understanding.”
Response 3: The text has been improved and a relationship with the parents' knowledge has been established.
P 2 l Introduction 49-55
Point 4:“The introduction does not give any rationale for the hypothesis that knowledge might be different in high complexity facilities rather than low complexity.”
Response 4: A rationale for our hypothesis an explanation about the motivation of the research has been added.
P 2 Introduction l 63-66
Point 5: “There is a high proportion of C-sections in both types of facility - is this usual or would more straightforward deliveries be homebirths? ie how does the population of women studied reflect women giving birth in the region?”
Response 5: These data on cesarean sections are for the sample as a whole and are contaminated by the high complexity group. Data on births in each group individually (high- and low-complexity) were added to the results.
P 5 Results l 196-198
In item 2.2 of the Methods section, the operational capacity of the high- and low-complexity facility and the flow followed by puerperal women in the local public health system were described.
P 2 Methods l 82-90
Point 6: “#2.3 there is high proportion of women claiming no knowledge of the topic. It would be helpful to explain how women get knowledge about screening as this is may be limited which would in turn limit the understanding of the participants. It would be useful also to suggest in the discussion how this might be improved (of course will be particular to the local healthcare system).”
Response 6: An explanation of the diversity in the information supply was included in item 2.3 and attached a reference (http://bvsms.saude.gov.br/bvs/publicacoes/triagem_neonatal_biologica_manual_tecnico.pdf) about technical aspects, didactic material developed by the Ministry of Health of Brazil.
P 3 Methods l 107-110
A proposal to increase the supply of information in prenatal care was also included in the discussion.
P8 Discussion l 292-295
Point 7: “l 166 number illiterate is missing.”
Response 7: Only one participant was illiterate. This information (n = 1, 0.5%) has been added to the text.
P 5 Results l 193-194
Point 8: “1 199-200 please clarify difference between diseases and most diagnoses.”
Response 8: The sentence has been improved.
P 5 Results l 228-230
Point 9: “l 206 and elsewhere. The usual meaning of indifferent is lack of interest or caring about a topic (although it does have secondary meanings around not having strong feelings), however neutral might be a better choice (meaning neither agreeing or disagreeing) as it does not have the implication of not caring.”
Response 9: We agree with the reviewer, and now use the term “neutral”.
Throughout the text
Point 10: “Discussion l 259-270 it is unclear how the difficulties of deciding what disorders are on screening panels and international and national inequity of screening relates to maternal knowledge in this community.”
Response 10: This passage has been removed.
Point 11: “l275-294 it is unclear how new genetic technologies and inadequacy of pain management impact women's knowledge of screening. The potential for screening by genomic methodologies is certainly important as are the limitations of this technology but the relevance to the study is not explained.”
Response 11: This passage has been removed.
Point 12: “l299 the newborn screening system is generally interpreted as being the entirety of actions which result in good outcomes for screened babies ie maternity care, birthing facilities, laboratory, paediatricians, family, rather than a panel of screened disorders.”
Response 12: We agree. A more appropriate conceptualization has been added to the text.
P 9 Discussion l 300-303
Point 13: “1305 and earlier - it would be interesting for the readers to know how non-paternity is discovered during the screening process (presumably only when family studies are done after a diagnosis is made) and whether separate informed consent is required at this stage (as it would be in many other places) ie not an important part of the initial stages of screening and diagnosis.”
Response 13: As this topic is not closely related to the initial stages of the screening test, it will be removed from the discussion, but will be kept in the Methods and Results.
P 9 Discussion 1 309-310
Point 14: “l309-312 is the most important part of the discussion and would be good to see positive suggestions.”
Response 14: A paragraph has been added containing such suggestions.
P 9 l 332-338
Point 15: “l317 recollection to confirm screening results is not generally considered an ethical issue any more than thyroid function tests and imaging ie part of a screening protocol.”
Response 15: This remark has been included in the text.
P 9 Discussion l 325
Point 16: “l325-327 would be good to see this part clarified.”
Response 16: The description was more for result than for discussion. The text has been removed.
Point 17: “An important aspect of this type of study is the impact on screening uptake of family knowledge. It would be relevant to know when the questionnaire was administered with respect to offering the test and whether the questionnaire affected uptake and how maternal knowledge (or lack of for the 39 excluded) affected screening acceptance for their babies.”
Response 17: This information was included in the limitations, as there was no intention within our initial planning of the study to verify the influence of the family as a repository of information, including in relation to newborn screening.
P 10 Discussion l 345-355
Point 18: “In summary the topic of maternal/family knowledge of bloodspot metabolic screening and how it impacts acceptance of both screening and followup/ diagnostic testing is critical in developing and refining effective newborn screening systems and if this write up was confined to issues around current newborn bloodspot testing and included suggestions for improvement and relevance to screening acceptance it would make a strong contribution to the field.”
Response 18: Part of the text was used to supplement our observations on family support.
Round 2
Reviewer 2 Report
This paper reads much more easily with the modifications although there are still some areas of doubtful relevance to the overall message unless the hope is that with more knowledge of newborn screening in the population there might be more pressure on health funders to add more tests to the programme and actively work to improve coverage in the region (in which case this might be stated), It remains unclear how representative these birth facilities are of the population of the region (the percent of C-section in the study population is given but at 54% this is high for all women giving birth - are there for example homebirths in the region?). Nevertheless the paper will be of interest to those in the screening community who are concerned about developing screening programmes.